# Sensory Features Introduced by Brewery Spent Grain with Impact on Consumers’ Motivations and Emotions for Fibre-Enriched Products

**DOI:** 10.3390/foods11010036

**Published:** 2021-12-24

**Authors:** Ana Curutchet, Maite Serantes, Carolina Pontet, Fatima Prisco, Patricia Arcia, Gabriel Barg, Juan Andres Menendez, Amparo Tárrega

**Affiliations:** 1Engineering Department, Catholic University of Uruguay, Montevideo 11600, Uruguay; ana.curutchet@ucu.edu.uy (A.C.); maite.serantes@correo.ucu.edu.uy (M.S.); caropontet96@gmail.com (C.P.); fatimapriscof@gmail.com (F.P.); 2Latitud-LATU Foundation, Av. Italia 6201, Montevideo 11500, Uruguay; 3Neuroscience and Learning Department, Catholic University of Uruguay, Montevideo 11600, Uruguay; gbarg@ucu.edu.uy (G.B.); JUAN.MENENDEZ@ucu.edu.uy (J.A.M.); 4Instituto de Agroquímica y Tecnología de Alimentos, CSIC, Avda. Agustín Escardino 7, 46980 Paterna, Spain; atarrega@iata.csic.es

**Keywords:** BSG, brewery, by-product, acceptability, valorisation, sensory

## Abstract

The aim of this work was both to formulate three different fibre-enriched products by the addition of Brewery Spent Grain (BSG), and to evaluate the impact of this fibre enrichment on sensory quality, acceptability, and purchase intention under blind conditions. BSG was incorporated into bread, pasta, and chocolate milk at levels of 8.3%, 2.8%, and 0.35% (*w*/*w*), respectively. The fibre-enriched products and their regular counterparts were evaluated together by consumers through a CATA questionnaire, the EsSense 25 Profile, an overall acceptability rating, and a purchase intention ranking. Although fibre-enriched bread and chocolate milk ranked lower in overall acceptability compared with their counterparts, no significant difference was found for fibre-enriched pasta (*p* > 0.05). Purchase intention did not differ significantly for both bread and pasta (*p* > 0.05), yet the reasons for purchasing them differed significantly (*p* < 0.05). Consumers recognised the fibre enrichment in these two products and, therefore, were willing to partially compromise on sensory attributes. The fibre-enriched chocolate milk, nonetheless, scored significantly (*p* < 0.05) lower in purchase intention than the control. This work demonstrates that the effect of BSG addition is product-specific, and that fibre perception makes consumers feel more confident.

## 1. Introduction

In recent years, extensive research has been undertaken to identify ingredients and components that display some clear physiological activity, namely improving intestinal transit [1,2] and the immunological system [3], or reducing cholesterol [4], blood glucose [5], arterial hypertension [6], and obesity [7].

The consumption of dietary fibre has potential benefits for human health, such as preventing chronic non-communicable diseases (NCDs) including obesity, diabetes, and cardiovascular disease [8,9]. Insoluble fibre prevents constipation, while soluble fibre is positively associated with decreased absorption of cholesterol and glucose at the intestinal level [10]. Indeed, a high dietary fibre intake has been correlated with a lower glycaemic response in diabetic patients [11].

Despite the evidence for the benefits of fibre intake, a worldwide fibre deficiency is observed in both developed and underdeveloped countries. A review by [12] revealed the daily average dietary fibre intake in adults for ten countries (including the USA, UK, and Spain), none of which reached the recommended amount. Although this recommendation varies depending on age and lifestyle, on average, it should be greater than 25 g/day according to the World Health Organization (WHO). 

In order to meet the demand of dietary fibre intake, fibre enrichment of food products has been widely proposed as a way to increase its consumption. This is especially appealing since NCDs are the leading causes of death in current times [13]. Among them, cardiovascular diseases are the most common NCD globally, leading to an estimated 17.8 million deaths in 2017 [14]. Furthermore, according to research published by the World Health Organization [13], the prevalence of obesity tripled between 1975 and 2016. 

Brewery Spent Grain (BSG) is a by-product obtained during the brewing process of malted barley, with an overall composition in dry matter of fibre (30–50%), protein (19–30%), lipids (10%), and ash (2–5%). The high amount of fibre as well as the high protein content in BSG justifies its use as a functional ingredient in the preparation of food for human consumption, and has resulted in research in the inclusion of BSG as a raw material. To mention some of them, [15] studied the valorisation of BSG through fibre-enriched egg pasta, reaching an optimised formulation of 6.2% of BSG. It has also been studied together with mushrooms, as an animal protein replacer and fibre-enricher of smoked sausages containing 3–6% of BSG [16]. Moreover, [17] obtained promising results for a snack chip containing 40% BSG inclusion. 

When formulating products with fibre, the fibre will likely impart a strong flavour, coarse texture, and dry mouthfeel [18]; therefore, consumers are sometimes reluctant to accept the reduction in sensory quality [19]. This is especially important if the product is mainly consumed for hedonic pleasure [20] as in the cases reported by [21,22], in which cakes and cookies, respectively, were enriched with fibre by adding fruit pomace and resulted in a reduction in overall acceptability and purchase intention. However, owing to the multiple health benefits derived from fibre enrichment, they can also be valued and consumed for their utilitarian nutritional value [20]. For instance, fibre enrichment did not have any perceivable effect on the overall acceptability of fermented pork sausages [23] and noodles [24]. According to [20], consumers’ responses may vary between hedonic and utilitarian products; therefore, two utilitarian products (bread and pasta) and one hedonic product (chocolate milk) were selected to study the effect of BSG enrichment.

The motivations driving food choice and tolerance vary from one food product to another. A hedonic scale is a simple method of measuring the overall liking scores, and it is considered the most informative assessment by consumers. However, consumer perception of a product goes beyond overall liking, also including extrinsic elements such as perceived benefits, quality, and wellness of the associated food. Additionally, cognitive–affective factors including perceived emotions, expectations, and health attitudes influence food choice [25,26]. Those factors produce a top-down modulation of food perception and attention that can modify consumer behaviour [27]. These extrinsic elements can be measured using advanced approaches, such as the EsSense Profile and check-all-that-apply (CATA) as conscious methods [28]. 

This work aimed to evaluate the use of BSG as fibre enrichment in three different product categories, namely bread, pasta, and chocolate milk, and the impact on consumer motivations and emotions that the fibre enrichment using BSG caused.

## 2. Materials and Methods

### 2.1. Brewers Spent Grains Flour Obtention

BSG (a mixture of *Hordeum vulgare* and *Zea mays)* was obtained from the Uruguayan brewing company Fábricas Nacionales de Cerveza S.A. BSG flour was produced by first drying the BSG, the same day it was generated, in a convection oven at 55 °C for 72 h followed by grounding it in a laboratory mill (Retsch ZM 200). The fraction that passed through a 1 mm mesh screen was used for the bread and pasta, while a 0.25 mm mesh screen was used for the chocolate milk. BSG flour was packed in polyethylene bags and kept at room temperature until use. For BSG composition analysis, the fraction that passed through a 1 mm mesh screen was used, except for fibre analysis for which a 0.5 mm mesh was used.

### 2.2. BSG Composition

Proximate analyses were performed on the BSG flour according to methods proposed by the Association of Official Analytical Collaboration [29]. Protein content was determined by Kjeldahl method and performed according to method 981.10 of the AOAC International with a conversion factor of 6.25. Fat content was determined by the soxtec method, as described in AOAC 2003.5. Quantification of moisture and ash content was performed under AOAC 925.09 and AOAC 923.02, respectively. Total dietary fibre was measured according to AOAC 985.29. Carbohydrates were calculated by difference (i.e., all other nutrients were summed and subtracted from the total weight of the sample). Sodium was estimated based on the values obtained by [30]. 

### 2.3. Product Formulation

Formulations for fibre-enriched bread, pasta and chocolate milk were developed. The amount of BSG flour used in each product was adjusted to reach a total fibre content of 2.5 g of dietary fibre per serving size, thus allowing the products to bear the “source of fibre” claim according to Uruguayan law [31]. This resulted in a content of dietary fibre of 2.5%, 5.0%, and 1.25% for pasta, bread and chocolate milk, respectively, which were used as the lower limit for BSG addition in the fibre-enriched products. These products also comply with the Codex Alimentarius [32] source of fibre claim, in particularly with the condition which calls for a minimum of 10% of the daily reference value per serving. BSG-enriched samples were fibre-enriched bread, fibre-enriched pasta and fibre-enriched chocolate milk, with BSG flour concentrations of 8.3%, 2.8% and 0.35%, respectively. Non-added-BSG formulations were also developed, by replacing the BSG with flour (in the case of bread and pasta) or milk (for the chocolate milk) in the fibre-enriched formulations, and used as control. These are the regular bread, regular pasta and regular chocolate milk. In Appendix A, the nutritional facts of each product are presented.

#### 2.3.1. Bread

Bread was formulated according to [30], and its formulation is detailed in Table 1. All ingredients were bought locally. The dough was formed in a KitchenAid professional Bowl-Lift Stand Mixer 5KSM7990X using a spiral dough hook, 6.9 L bowl and a low-speed setting. For the fibre-enriched bread, dry ingredients were first mixed with water for 15 min. Oil and BSG flour (<1 mm) were then added in turns and mixed for 5 min after each addition. The regular bread dough was made likewise, except for the final mixing, which was omitted. Once the doughs were formed, they were left to bulk ferment at 35 °C and 60% relative humidity (RH) for 30 min. Doughs were shaped in 27.5 × 15.0 × 6.5 cm^3^ (L × W × H) loaf pans and left to further ferment for 50 min under the same conditions. The proofed breads were baked at 190 °C for 30 min in a convection oven. After baking, they were cooled for 20 min, packed in sealed polyethylene bags, and stored refrigerated until sensory evaluation took place 24 h later. 

#### 2.3.2. Pasta

The ingredients for the pasta were: wheat flour, BSG flour (<1 mm), water and eggs. The formulation was adapted from the pasta maker machine recipe book (Table 2). All ingredients were bought at a local supermarket. Fresh pasta was made with a pasta maker machine (HR2355/08, Philips, Amsterdam, The Netherlands), using the standard program and following its instructions. Spaghetti were cut at a length of approximately 30 cm and left at room temperature until the moment of cooking. The pasta was always consumed the same day that it was made.

#### 2.3.3. Chocolate Milk

The regular chocolate milk was formulated according to a locally commercial chocolate milk, to which BSG flour was then blended in to create the fibre-enriched version. Chocolate milk was made in a mixer, by mixing semi-skimmed milk (1.5% fat content), BSG flour (<250 μm), cocoa powder (12% fat content), granulated sugar, carrageenan type A, carboxymethyl cellulose (CMC) and Givaudan chocolate flavouring for 2 min. Formulations are detailed in Table 3. All ingredients were bought locally.

### 2.4. Consumers’ Evaluation Sessions

To study the acceptability of products, one session was held for each product, under blind conditions and on different days. The number of consumers that participated was 110, 119 and 111 in bread, pasta and chocolate milk product sessions, respectively. Participants were randomly recruited on the university campus and public places, being regular consumers of each product between June and September of 2021. The panellists, male and female, ranged between 18 and 70 years old. For the sensory evaluation, the participants were provided with two samples—one of the regular product and one of the fibre-enriched product. These samples were given without any information related to their composition or health effects. Water was provided to consumers as a palate cleanser between samples. 

#### 2.4.1. Bread

Breads were prepared the previous day and refrigerated in plastic bags to avoid moisture loss. On the day of evaluation, breads were removed from the refrigerator and left to warm up to room temperature. Half slices of fibre-enriched bread and regular bread (approximately 15 g) were served in biodegradable plastic plates coded with a three-digit number. 

#### 2.4.2. Pasta

Fresh pasta was made on the day of evaluation. BSG-added and regular pasta were cooked in abundant salted (1 tbl. spoon) water for 15 min, drained and seasoned with extra virgin olive oil (1 tbl. spoon). Approximately 15 g of cooked spaghetti was served in white polystyrene cups coded with a three-digit number. Samples were maintained warm until evaluation.

#### 2.4.3. Chocolate Milk

Chocolate milk samples were prepared, refrigerated and evaluated in less than 24 h after being produced. Samples were served at 5 °C in plastic cups coded with a three-digit number.

### 2.5. Sensory Evaluation Questionnaire

Three-digit-coded samples of both products were randomly presented to regular consumers of each product. Participants completed a computer-based questionnaire. The questionnaire consisted of two sections. Section one measured consumers’ perception of the product and section two determined the emotions provoked by the product in consumers.

In section one, participants were asked to rate the overall acceptability using a nine-point hedonic scale (1—extremely disliked to 9—extremely liked). Subsequently, consumers completed a CATA product-specific questionnaire to describe the sample. CATA attributes were determined through preliminary brainstorming sessions with consumers who tasted similar products. The attributes were randomly presented to participants to avoid order effects. Finally, purchase intention was measured by a five-point scale from 1—would definitely not buy to 5—would definitely buy, followed by a CATA questionnaire to uncover the reason behind their decision (Table 6).

In section two, the EsSense25 questionnaire was used to collect emotional response data based on the methodology suggested by [33]. The twenty-five emotions were first translated into Spanish by the authors. As Spanish and Portuguese are closely related due to their Latin roots, Spanish translations were then compared to the Portuguese terms suggested by [34] and then back-translated to English. Emotions in the EsSense 25 profile are classified into positive, negative and unclassified subgroups according to their nature [35]. Consumers were presented with a five-point intensity scale (1—Nothing, 2—Slightly, 3—Mildly, 4—Very and 5—Extremely) and asked to select the most accurate intensity for every one of the twenty-five emotions. The emotions were randomly presented to participants to avoid order effects. See Appendix A for the full questionnaire design.

### 2.6. Data Analysis

Acceptability, purchase intention and EsSense 25 results obtained were subjected to paired samples Student’s *t*-test analysis with a significance level of 0.05. Raw CATA data were converted into binary data based on frequency of citation (1—cited, 0—non-cited). Nonparametric Cochran’s Q test was then performed on the binary CATA data to detect significant differences (*p* < 0.05) among fibre-enriched and regular samples for each attribute. Based on recent research published by [35], EsSense25 results were subjected to paired samples Student’s *t*-test analyses at a significance level of 0.05 to investigate difference in the intensity of emotions between fibre-enriched products and their counterparts. All statistical analysis was performed using XLSTAT software (2019 version, Addinsoft Inc., New York, NY, USA).

## 3. Results and Discussion

### 3.1. BSG Flour Composition

The BSG flour composition is shown in Table 4, including protein, fat, carbohydrates, dietary fibre, ash, sodium and moisture contents. Results show a notable feature: the total dietary fibre content is higher than the one obtained by [30], which was 44.61 ± 0.29%. General composition is in the reported range [36]. The significant amount of this nutrient allows the formulation of products with the addition of BSG as a functional ingredient, and for them to bear the “source of fibre” claim.

### 3.2. Hedonic Analysis and Emotional Profile

#### 3.2.1. Bread

The acceptability differed significantly for regular and fibre-enriched bread (Table 5). Similar results were obtained by [30,37], where the BSG addition to bread dropped consumers overall acceptability. On the other hand, for purchase intention, no significant difference was found. Consumers may have identified the health benefits of fibre-enriched bread based on its appearance, even though no information was given [38]. Awareness of health benefits may have caused consumers to compromise on the product’s sensory features. Indeed, for utilitarian products such as bread, sensory quality is not the main driver of consumer choice, as opposed to hedonic products [20].

The most frequently mentioned reasons for consumers choosing to buy the fibre-enriched bread (67%) were that it was healthy (64%), tasty (81%), to include fibre in their diet (57%) and to avoid weight gain (19%) (Table 6). Except for being tasty, and because their kids would like it, all the other reasons were more frequently selected for fibre-enriched bread than for regular bread (*p* < 0.05). Despite the high frequency of choice of the tasty attribute for fibre-enriched bread, it was more commonly selected as a reason for buying the regular bread (*p* < 0.05), which accounted for a vast 98% of the consumers willing to buy the latter. 

Despite what [20] found, for both fibre-enriched and regular bread, the main reason for not buying them was related to their taste, where fibre-enriched bread ranked higher than regular bread (*p* < 0.05): 61% for fibre-enriched compared to 33% for regular bread (Table 6). Sensory properties have already been reported to be the main reason for not consuming a fibre-enriched product [39]. Unlike for regular bread, price resulted in a relatively common reason for not buying the fibre-enriched one (*p* < 0.05). This is supported by [40], which states expensiveness as a misperception consumers have towards fibre-enriched products. Furthermore, the reason related to consumers’ families not liking this type of product may be due to the low popularity of fibre-enriched products in children and adolescents [41,42]. 

With respect to the CATA questionnaire, 12 out of the 21 attributes detailed showed a significant difference in terms of frequency of mention (Figure 1). The attributes most used to describe fibre-enriched bread were fibrous, natural, soft, compact, fluffy, and dry. For regular bread, soft, fluffy, natural, and moist were the attributes most frequently selected. Despite many of these attributes being recurrent in both bread samples, repeated frequent attributes, except for natural, differed significantly on the frequency of mention. Fibre-enriched breads have already been reported to have a decreased specific volume compared to non-enriched [37,43]. Additionally, consumers may have described the fibre-enriched bread as dry since fibre has a high-water holding capacity [44] and free water may seem scarce in this type of bread. 

Five of the twenty-five emotional terms showed significant difference between bread samples (Table 7). Two were positive emotions—warm (*p* = 0.000) and active (*p* = 0.027)—with participants being more active after testing the regular bread but less warm. Additionally, tame (*p* = 0.001) and guilty (*p* = 0.005) showed a significant difference. Participants felt more guilty and less tame after testing the regular bread. Finally, respondents felt significantly more bored (*p* = 0.000) when consuming the fibre-enriched bread.

#### 3.2.2. Pasta

For pasta, no significant differences were found in either acceptability or purchase intention between the regular and the fibre-enriched product (Table 6). This first result is in accordance with [39], who did not find a significant difference between the liking of traditional pasta and the liking of 10% wheat-bran-added pasta. The fact that consumers were equally willing to buy regular and fibre-enriched pasta is understandable based on the previous result on liking. This was an interesting finding, as only a 15.8% of the total grains consumed on a given day are whole grains, according to the U.S. Department of Health and Human Services [45]. It may thus seem that respondents were not against fibre-enriched pasta, but unaware of it as a replacement to regular pasta [39,46]. Indeed, [46] reported that their focus group participants chose whole grain breads more regularly than whole grain pasta, for which they usually opted for the refined version. 

In this case, 76% of consumers were willing to buy the fibre-enriched pasta. The most frequently given reasons for buying it were: because it is tasty (86.7%), healthy (58.9%) and because they want to include fibre in their diets (3.7%) (Table 6). The main reason for buying regular pasta was that it is tasty (94.1%), while all the other reasons were negligible in terms of frequency of mention (<20%). Significant differences (*p* < 0.05) were detected in the frequency of mention of healthy and to include fibre in their diet between the two products. 

On the other hand, the main reasons for not buying fibre-enriched pasta were because it is distasteful (66.7%) and because is not the one they always buy (48.1%) (Table 6). The first one is once again related to fibre imparting a strong flavour, coarse texture and dry mouthfeel, as argued for bread samples [18]. In addition, the last reason can be associated with food neophobia, which is defined as the reluctance to eat new foods [47] and has already been reported to be a barrier to novel foods consumption [48].

The CATA questionnaire results for pasta showed that 11 out of the 20 terms significantly differed in their frequency of choice between samples (Figure 2). Participants associated the fibre-enriched pasta with the attributes: natural, fibrous, granulated, coarse, dark, gritty and strange taste. On the other hand, the terms soft, uniform, homogeneous, and flexible were associated with regular pasta. Moreover, both pastas were frequently described as tasty with no significant difference between them.

Numerous texture attributes differed significantly in frequency of mention between both samples, suggesting that addition of BSG to pasta produces textural and flavour changes. This is not surprising, as this is similar to results obtained by [49], who studied whole barley incorporation into regular pasta for a total final dietary fibre concentration of 2.6% (*w*/*w*). Their results showed that both firmness and adhesiveness decreased with the amount of barley being added, whilst taste and overall liking ranked significantly lower for barley-added pasta when compared to the control sample. This is apparent by the results of CATA attributes—fibre-enriched pasta was described as more fibrous, granulated, coarse and gritty and less soft, uniform, homogenous and flexible compared to the regular sample. Certainly, the addition of fibre interferes with gluten network and prevents the cohesiveness of the dough. As a result, starch granules are more easily lost during cooking and the quality of the final product decreases [49]. It should be noted that a possible solution to these negative effects may be the addition of xanthan gum and vital gluten. In [50], equal texture and flavour characteristics for both the 5% (*w*/*w*) β-glucans-added pasta and control sample were obtained, by adding 5% xanthan gum and vital gluten.

Moreover, the colour difference detected between samples is due to BSG flour being darker than wheat flour, as already observed by [51] in BSG-enriched pasta. A decrease in yellowness and increase in brownness and redness was observed for all 5%, 10% and 20% (*w*/*w*) BSG addition levels in dried pasta. This may have decreased its acceptability as the yellow colour is highly valued by consumers of durum wheat pasta [51].

Fourteen of the twenty-five emotional terms presented significant differences in their rating when comparing fibre-enriched and regular pasta (Table 7), indicating that the two products were perceived in very different ways. Two negative emotional terms—bored (*p* < 0.0001) and disgusted (*p* < 0.0001)—presented significant differences. While the first term was significantly more elicited for the BSG-added sample, the opposite occurred for the latter. Other emotions that presented significant difference included: guilty (*p* < 0.0001) and wild (*p* = 0.036), which were less frequently applied to fibre-enriched pasta when compared to regular pasta, and tame (*p* < 0.0001), which was rated higher for the fibre-enriched version.

#### 3.2.3. Chocolate Milk

Significant differences were found in both the acceptability and purchase intention of chocolate milk products (Table 6). These differences were probably rooted in the fact that fibre inclusion introduces changes in the texture and flavour of the product, as evidenced by the results of CATA attributes and reasons for not purchasing the product. Among consumers willing to buy fibre-enriched chocolate milk (39.6%), most of them (95.5%) stated that taste was the main driver for purchasing (Table 6). However, this was also the main cited reason for buying regular chocolate milk, scoring a significantly higher frequency of mention compared to the first one (*p* < 0.05). This difference in frequency of choice is mostly due to the higher percentage of consumers buying the regular sample (60.4%). 

In addition, 39.6% of consumers were willing to buy the fibre-enriched chocolate milk. The main reason given against buying it (60.4% of consumers) was due to its taste, this being more frequently selected than for the regular product (*p* < 0.05) (Table 6). This attribute was also chosen for bread and pasta as main reasons for not buying the products. In [52], it was reported that the incorporation of BSG into baked snacks altered the flavour profile of the snacks, in addition to their taste and overall acceptability. Researchers associated this unpleasant odour with compounds deriving from fermentation and the Maillard reaction, including 3-methyl-butanal, 2,3 butanedione and 2-methy-butanal, which were present in high quantities in both BSG flour and the BSG snacks.

For chocolate milk, 7 of the 20 terms presented in the CATA questionnaire demonstrated significant differences (*p* < 0.05) (Figure 3). Both chocolate milks were frequently associated with the attributes smooth, strong chocolate flavour and natural. However, the attributes gritty, artificial, coarse, aftertaste, strange taste, off-flavour and greasy distinguished the fibre-enriched chocolate milk from the regular one (*p* < 0.05). Those attributes may explain the significant lower acceptability obtained for the BSG-added product, affecting both product texture and flavour. 

The acceptability of fibre-enriched milk has not been thoroughly studied in the literature. However, its addition to other fluid dairy products, such as yoghurt, has been investigated. Particularly, [53] evaluated the sensory characteristics and consumer acceptance of fibre-enriched yoghurts, reporting a significantly lower acceptability compared to the control sample, caused primarily by a gritty or sandy texture. This conclusion is in accordance with the results obtained in this study, where fibre-enriched chocolate milk was significantly less accepted than the regular one and the gritty attribute was selected by 30% of participants for the BSG product (12% for the regular chocolate milk). In [54], the perceived oral grittiness of multiparticulate formulations was studied, and it positively correlated with the amount and size of particles being dispersed in the media. The product matrix has also been reported to greatly influence the particle size detection threshold, concretely particles being dispersed in liquid matrices are easier to detect than the ones in solid matrices [55]. Although the BSG particle size used for fibre-enriched chocolate milk was smaller than for fibre-enriched bread and pasta, it was still considerably large (>200 μm) compared to detection thresholds for cellulose particles (1.5% *w*/*w*) in viscous low-fat quark and semi-solid high-fat processed cheese, which are estimated to be 52 μm and 82 μm, respectively [55]. In addition, the product matrix being liquid is likely to have facilitated the perception of grittiness compared to semi-solid or solid foods. For all the above, unpleasant roughness and grittiness was still perceived, albeit BSG concentration being considerably low (0.35% *w*/*w*).

Furthermore, although bitterness was relatively more frequently used to describe the fibre-enriched chocolate milk, its frequency of mention did not differ significantly between samples (*p* < 0.05). In [53], it was reported that bitterness had a negative influence in the flavour quality scores of 30 g/kg fibre-enriched yoghurts, which were significantly lower than for the 15 g/kg counterparts. This result suggests that fibre addition may have increased the bitterness of the fibre-enriched chocolate milk, yet its concentration was not high enough to exert a significant effect. 

Just as taste scored significantly higher in terms of frequency of mention among the reasons for not buying fibre-enriched chocolate milk compared to the regular one (*p* < 0.05), significant differences (*p* < 0.05) were also found in frequency of mention of the following CATA attributes describing taste: unpleasant aftertaste, off-flavour and strange taste. Again, this shows the unpleasant and negative effects of incorporating BSG into foods [52]. Nonetheless, a decrease in sensorial properties (particularly aroma and flavour) has also been reported after incorporating cauliflower by-product extracts into apple juice [56], becoming apparent it is one of the greatest struggles of by-products’ valorisation. 

In the case of chocolate milk, three emotional terms showed significant difference in their mean scores between samples (Table 7): calm (*p* = 0.016), mild (*p* = 0.001) and tame (*p* = 0.002). Participants felt significantly milder after consuming the fibre-enriched chocolate milk, yet they felt tamer after testing the regular chocolate milk. A positive emotion also presented significant differences in its rating—calm with *p* = 0.016—this emotion being more frequently elicited for the fibre-enriched chocolate milk.

### 3.3. Comparison of the Fibre Impact on Different Category of Products 

For different food product categories, consumers hold specific sensory and hedonic expectations which subsequently affect their food perception and acceptance [57]. In fact, previous experiences, product characteristics and context shape expectations. Sensory attributes of the product must meet consumer expectations for the acceptability to be high. In this work, consumers responded differently to the fibre enrichment of bread, pasta and chocolate milk. Interestingly, consumer responses towards bread and pasta were remarkably similar. In neither products consumer showed a statistically difference in purchase intention between the regular and fibre-enriched version. Amidst the reasons for buying the fibre-enriched bread and pasta, consumers cited healthy and to include fibre in their diets. These results confirm the ability of respondents to successfully recognise fibre-enriched products and, thus, infer their health benefits. Knowledge about dietary fibre and its health effects has already been reported to be substantial among the general population [58]. In [59], it was reported that information on fibre displayed on labels was well-understood by consumers and gave a positive preference for high-fibre products.

In the case of chocolate milk, consumers could not recognise the addition of fibre to the product, and instead, perceived a strong and unexpected flavour, coming from BSG’s strong aromatic compounds, enhanced due to the product being liquid. Although some respondents mention the nutty flavour characteristic of the fibre-enriched version, most did not, as shown by the low acceptability and purchase intention rating. Lack of awareness of its nutritional benefits may have prevented the “halo” effect, which is defined as a positive influence on the perception of a product due to an unrelated impression from another attribute [60,61]. Indeed, communication of a product’s health benefits can be a powerful marketing tool for the food sector due to its potential “halo” effect. For instance, [21] reported an increase in liking for fibre-enriched apple pomace cake when the claim source of fibre was displayed, showing the impact, or “halo” effect, of health claims on hedonic liking. Thus, although fibre-enriched chocolate milk was not liked as much as the control, including the corresponding nutritional claims may bridge the gap between them. Pasta is the product for which emotional response differed the most, despite being the only one with no significant difference in either acceptability or purchase intention. Differences in emotional response between regular and BSG-enriched samples were evoked due to differences in their sensory features, as no information was provided. Five emotions were repeated in at least two categories and show significant differences between the fibre-enriched and the regular one: tame, calm, warm, bored, guilty and active. 

Only tame had the same behaviour in the three food categories; certainly, fibre enrichment made consumers feel more tame. However, although consumers felt more warm, bored, active and less guilty when tasting fibre-enriched pasta and bread, the opposite occurred for fibre-enriched chocolate milk. This seems to indicate that the knowledge or perception of fibre enrichment provokes a distinct set of emotions. In [62], the emotions evoked by beer consumption in blind and informed conditions were studied, finding that the label information provided to participants increased their acceptance, induced some positive emotions and increased their willingness to pay (WTP) for pure malt beers.

Emotions reported by consumers confirmed a top-down modulation produced by previous experiences related to fibre content. For both bread and pasta, sensory qualities derived from fibre addition were perceived as healthy. A top-down modulation is observed in the high score that the statement “because it is healthy” obtained as a motivation to purchase these products (Table 6). This is also seen in the emotional response. Fibre-enriched bread and pasta evoked emotions associated with health (warm, bored, tame, active), while regular bread and pasta induced guilt, an emotion associated with highly caloric non-healthy food intake, as was described by [63,64]. This pattern was not present in the chocolate milk, as chocolate is strongly associated with hedonic expectations and the fibre inclusion modified the standard characteristics of the product.

## 4. Conclusions

In this work, the effect of BSG addition to different food product categories was analysed by sensory evaluations of both control and fibre-enriched samples. The amount of BSG flour added to the fibre-enriched products was enough for them to bear the claim Source of fibre according to Codex Alimentarius. The addition of fibre led to some changes in acceptability and buying intention as well as in product perception. Results showed that BSG enrichment had a significant effect on the sensory properties of all three products (*p* < 0.05), affecting both texture and flavour. Although the fibre-enriched bread and fibre-enriched chocolate milk ranked lower in overall acceptability compared with their counterparts, no significant difference was found between fibre-enriched pasta and regular pasta (*p* > 0.05).

Interestingly, purchase intention did not differ significantly for either bread or pasta (*p* > 0.05), yet the reasons for purchasing them differed significantly (*p* < 0.05). It seems consumers were aware of fibre-enrichment in these two, and were willing to partially compromise on sensory attributes. Fibre-enriched chocolate milk, nonetheless, scored significantly lower in purchase intention than the regular product, probably due to the mouthfeel perception being affected by BSG particles and lack of awareness of it being a functional product. 

Overall, the findings demonstrated that the effect of BSG addition is product-specific; while fibre-enriched pasta and fibre-enriched bread were approved by consumers, results for fibre-enriched chocolate milk suggested the opposite. Ambivalence was seen in the emotions generated by the different fibre enriched products, depending on whether the fibre was perceived or not. When consumers perceived the fibre enrichment in the product, they approved consumption, feeling in general more confident. 

Finally, it should be noted that this study used convenience sampling of consumers and only three products were investigated. Further research considering the demographic characteristics of the consumers and including other products should be conducted to determine the full effect of fibre enrichment with BSG in consumers’ response. Future research should also focus on the effect of information on the emotions provoked by fibre-enriched products, and their effects on attention, to draw more in-depth conclusions.

## Figures and Tables

**Figure 1 foods-11-00036-f001:**
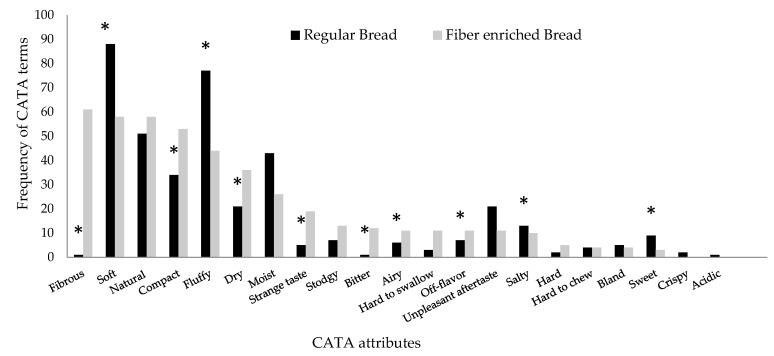
Frequency of CATA terms for fibre-enriched bread and regular bread. * indicates significant difference at *p* < 0.05.

**Figure 2 foods-11-00036-f002:**
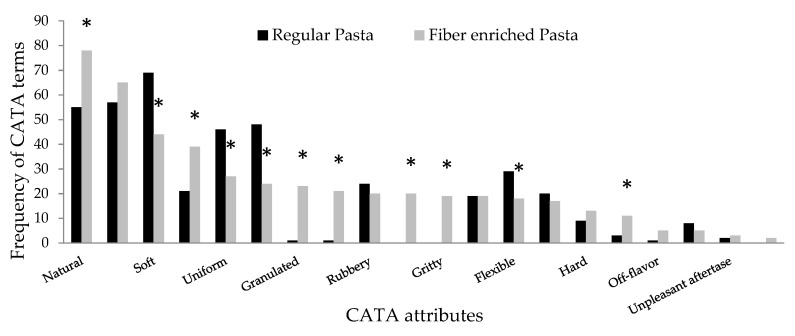
Frequency of CATA terms for fibre-enriched pasta and regular pasta. * indicates significant difference at *p* < 0.05.

**Figure 3 foods-11-00036-f003:**
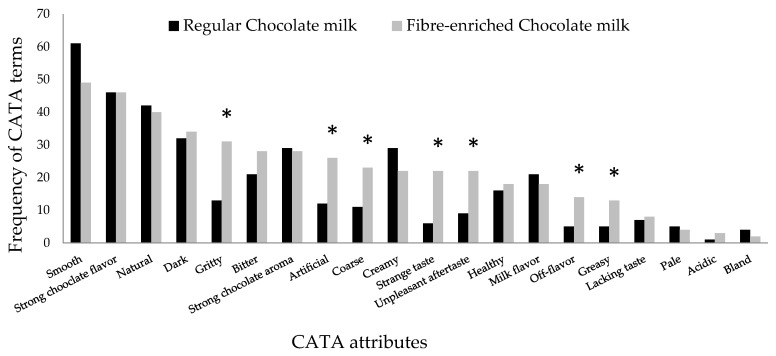
Frequency of CATA terms for fibre-enriched chocolate milk and regular chocolate milk. * indicates significant difference at *p* < 0.05.

**Table 1 foods-11-00036-t001:** Formulations of fibre-enriched bread and regular bread.

Ingredients	Fibre-Enriched Bread (%)	Regular Bread (%)
Water	33.37	33.37
Wheat flour	45.00	53.30
BSG flour (<1 mm)	8.30	-
Sunflower oil	4.86	4.86
Granulated sugar	3.60	3.60
Skim milk powder	3.20	3.20
Salt	1.20	1.20
Instant yeast	0.47	0.47

**Table 2 foods-11-00036-t002:** Formulations of fibre-enriched pasta and regular pasta.

Ingredients	Fibre-Enriched Pasta (%)	Regular Pasta (%)
Water	11.97	11.97
Wheat flour	70.06	72.86
BSG flour (<1 mm)	2.80	-
Egg	15.97	15.97

**Table 3 foods-11-00036-t003:** Formulations of the fibre-enriched chocolate milk and regular chocolate milk.

Ingredients	Fibre-Enriched Chocolate Milk (%)	Regular Chocolate Milk (%)
Semi-skim milk	89.47	89.82
Granulated sugar	7.00	7.00
Cacao powder	3.00	3.00
BSG flour (<250 μm)	0.35	-
Carrageenan Type A	0.075	0.075
CMC	0.075	0.075
Chocolate flavouring	0.03	0.03

**Table 4 foods-11-00036-t004:** BSG flour composition per 100 g.

Component	Mean Value ± SD
Proteins (g)	24.9 ± 0.02
Lipids (g)	6.28 ± 0.16
Carbohydrates (g)	16.24 *
Total dietary fibre (g)	45.47 ± 0.34
Ash (g)	2.81 ± 0.25
Sodium (mg)	240
Moisture (g)	4.3 ± 0.4

* Carbohydrates were determined by difference.

**Table 5 foods-11-00036-t005:** Acceptability and purchase intention for bread, pasta and chocolate milk; regular and fibre-enriched product.

Product	Acceptability *	Purchase Intention **
** *Bread* **		
Regular bread	7.10 ± 1.48 ^a^	3.93 ± 1.00 ^a^
Fibre-enriched bread	6.44 ± 2.06 ^b^	3.75 ± 1.26 ^a^
** *Pasta* **		
Regular pasta	6.87 ± 1.45 ^a^	4.14 ± 0.93 ^a^
Fibre-enriched pasta	6.91 ± 1.91 ^a^	4.00 ± 1.20 ^a^
** *Chocolate milk* **		
Regular chocolate milk	6.47 ± 1.86 ^a^	3.55 ± 1.26 ^a^
Fibre-enriched chocolate milk	5.68 ± 2.13 ^b^	3.03 ± 1.34 ^b^

Acceptability and Purchase intention expressed as mean ± SD. For each category, product scores not sharing letters are significantly different (*p* < 0.05) according to Student’s *t*-test. * Evaluated in a structured nine-point scale. ** Evaluated in a structured five-point scale.

**Table 6 foods-11-00036-t006:** Reasons for buying and not buying the regular and fibre-enriched samples of bread, pasta and chocolate milk.

	Frequency of Citation (%)					
	Bread		Pasta		Chocolate Milk	
	Regular	Fibre-enriched	Regular	Fibre-enriched	Regular	Fibre-enriched
** *Reasons to buy* **						
Because it is healthy	**21.3**	**63.5**	**23.8**	**58.9**	13.4	20.5
Because it is tasty	**97.5**	**81.1**	**94.1**	**86.7**	**97.0**	**95.5**
To avoid weight gain	**1.3**	**18.9**	3.0	11.1	0.0	0.0
Because it is high in calories	2.5	1.4	4.0	4.4	4.5	6.8
Because it is good for my family	10.0	20.3	9.9	13.3	4.5	9.1
Because I want to include fibre in my diet	**11.3**	**56.8**	**14.9**	**36.7**	11.9	15.9
Because my kids would like it	**16.3**	**4.1**	12.9	7.8	9.0	15.9
** *Reasons not to buy* **						
Because it does not seem healthy	30.0	8.3	**35.3**	**3.7**	11.4	13.4
Because it is distasteful	**33.3**	**61.1**	**29.4**	**66.7**	**52.3**	**56.7**
Because it is not the one I always buy	10.0	22.2	**17.6**	**48.1**	**18.2**	**26.9**
Because I do not need fibre	3.3	0.0	5.9	0.0	4.5	4.5
To prevent weight gain	20.0	19.4	23.5	14.8	15.9	14.9
Because I think it is bad for my family	0.0	0.0	17.6	3.7	2.3	1.5
Because it seems expensive	**0.0**	**11.1**	**0.0**	**18.5**	6.8	1.5
Because I don’t want to include fibre in my diet	6.7	5.6	17.6	3.7	2.3	3.0
Because my family would not like it	**6.7**	**25.0**	23.5	18.5	4.5	10.4

Values in bold indicate statistically significant differences (*p* < 0.05) between samples.

**Table 7 foods-11-00036-t007:** Effect of product on consumers’ emotional response for bread, pasta and chocolate milk.

Mean Scores	Bread			Pasta			Chocolate Milk		
	Regular	Fibre-Enriched	*p* Value	Regular	Fibre-Enriched	*p* Value	Regular	Fibre-Enriched	*p* Value
Active (+)	**3.02**	**2.66**	**0.027**	**2.85**	**2.50**	**0.021**	2.67	2.64	0.872
Adventurous (+)	2.44	2.56	0.458	2.60	2.86	0.109	2.39	2.31	0.632
Aggressive	1.92	1.86	0.693	2.33	2.30	0.875	1.62	1.62	0.956
Bored (−)	**1.66**	**2.22**	**0.0001**	**1.54**	**2.70**	**<0.0001**	2.18	1.87	0.052
Calm (+)	2.57	2.81	0.123	**1.93**	**2.86**	**<0.0001**	**2.41**	**2.81**	**0.016**
Disgusted (−)	1.69	1.56	0.353	**2.37**	**1.67**	**<0.0001**	1.80	1.87	0.742
Enthusiastic (+)	2.98	2.97	0.947	2.99	3.08	0.512	2.71	2.70	0.931
Free (+)	2.8	2.59	0.227	**2.69**	**2.10**	**0.0003**	2.58	2.73	0.415
Good (+)	2.97	2.99	0.914	2.59	2.49	0.549	2.79	3.01	0.178
Good natured (+)	2.97	2.45	0.129	**2.92**	**2.32**	**0.0001**	2.50	2.67	0.267
Guilty	**1.93**	**1.52**	**0.005**	**2.28**	**1.58**	**<0.0001**	1.45	1.67	0.083
Happy (+)	3.07	2.85	0.187	**2.94**	**2.53**	**0.011**	2.71	3.01	0.070
Interested (+)	2.93	3.14	0.170	2.95	3.05	0.464	2.64	2.66	0.889
Joyful (+)	2.84	3.03	0.246	**2.54**	**2.99**	**0.003**	2.95	2.78	0.283
Loving (+)	2.45	2.46	0.931	**2.15**	**2.52**	**0.015**	2.44	2.49	0.744
Mild	2.18	2.16	0.885	2.42	2.27	0.339	**1.65**	**2.08**	**0.001**
Nostalgic (+)	2.36	2.19	0.298	2.67	2.72	0.782	2.37	2.36	0.988
Pleasant (+)	2.66	2.45	0.170	**2.48**	**2.13**	**0.029**	2.99	2.71	0.071
Satisfied (+)	2.88	2.71	0.310	2.53	2.24	0.079	2.97	2.92	0.716
Secure (+)	2.9	2.89	0.976	3.10	3.00	0.478	2.58	2.80	0.211
Tame	**1.6**	**2.1**	**0.001**	**1.69**	**2.55**	**<0.0001**	**2.02**	**1.57**	**0.002**
Understanding (+)	3.02	2.89	0.360	3.04	2.94	0.499	2.87	2.80	0.669
Warm (+)	**2.29**	**2.88**	**0.0001**	**2.25**	**3.05**	**<0.0001**	2.78	2.49	0.095
Wild	2.25	2.15	0.567	**2.64**	**2.30**	**0.036**	1.97	2.06	0.583
Worried (-)	2.17	2.29	0.437	**2.67**	**2.67**	**0.979**	2.33	2.22	0.503

Classification of terms described by EsSense profile in positive (+), negative (−) or unclassified. Emotions in bold are significant difference between regular and fibre-enriched version.

## Data Availability

The datasets generated for this study are available on request to the corresponding author.

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
