# Peer review of "Sensory Features Introduced by Brewery Spent Grain with Impact on Consumers’ Motivations and Emotions for Fibre-Enriched Products"

_foods, 2021, doi:10.3390/foods11010036_

Round 1

Reviewer 1 Report

Although the study has some interesting elements and the topic is current,  it has major shortcomings.

First of all, the manuscript needs to be written in more scientific manner. Serious revision in the English language is highly recommended, as well.

In the present form, the concept of paper is not clear enough. The flow of writting from one sentence to the next should be smooth and in accordance with subheadings which is not a case in this paper. Serious modification of subheadings content is also necessary, particularly for the following:

  1. Introduction section: please rewrite this section in the manner to cover recent scientific studies relevant to the investigated topic
  2. Material and methods: in the present form it is not clear enough, does not provide sufficient information; Lines 107 - 111 should be deleted or move to discussion section; Lines 112-121: This part is very confusing and additionally burden text. It is correct to emphasize the regulations that you were guided by when creating formulations, however, it is necessary to provide a table with the chemical composition of the final products which will confirm that products really posses higher content of fibers.  Lines 121-123: please use more appropriate term for samples identification (suggestion Control B, Control P, Control CM instead of Regular, the same is for enriched samples); Line 135: 20 minutes after baking is not enough for stabilization and cooling of bread; please use SI units (24 hs?); Lines 161-169: Please improve this part. How did you choose consumers for the evaluation of samples? How often they consume these types of products? Please provide age range for every consumer groups that participated in the evaluation of particular product. Please clarify did you request Ethical Committee approval for performing sensory analysis with consumers? Lines 187-199: This part should be extensively rewritten. Please provide terms used for CATA and EsSense25 profile   
  3. Results and discussion: The results and discussion section is poorly written. Some of the conclusions were drawn arbitrarily and did not in accordance with the presented results (for example lines 238-242).  Lines 223-234: Please delete this part. The importance of BSG flour composition is not relevant for the present study. It is much more important to provide chemical composition of the final products.  

Some parts of the text are not logically arranged, while the sentences are not connected by transitions, which confuse readers. So I warmly recommend to completely rearrange the paper accordingly, keeping the focus on the main idea.

Author Response

Thank you very much for the feedback and for taking time to review the work carefully. The manuscript was deeply revised including the three reviewers' comments.  

Here are changes we made in order to accommodate to your comments: 

Introduction section 

In order to improve the flow of writing, some sentences were added to make transitions smoother. 

Most of the references used in the introduction section are from the last five years. 

Material and methods 

  • Lines 107 – 111. Sentence was modified.  
  • We considered that the chemical compositions of the final products are not part of the aims of the work. We needed to determine the fibre content of the raw material (BSG) and this data was obtained. What is important to the work is that the three products formulations reach the source of fibre claim according to EU legislation (codex!!). 
  • We considered that the sample identification used in this work allows readers to easily identify the products. 
  •  Hs was replaced by hrs. All units were checked. 
  • Consumer evaluation sessions was rewrittenan ethical statement is included at the end of the manuscript. 
  • Full questionnaire for each product was included in Appendix. There can be seen the terms used for CATA and EsSense25 profile    

Results and discussion:  

  • Lines 238-242 was rewritten. 
  • We think BSG composition is important to show is an ingredient useful for fibre enrichment, and to calculate with the three product formulations that the fiber enriched products are source of fibre. 

Reviewer 2 Report

The present manuscript shows an interesting study on the enrichment of products with Brewery Spent Grain. I have included some line by line recommendations and feedback in the attached document.

Author Response

Thank you very much for the positive feedback and for taking time to review the paper carefully. Here are changes we made in order to your comments: 

Abstract 

  • We include BSG definition in abstract in Line 17. 
  • A concluding sentence was included in the end of the abstract. 

Introduction 

  • Suggestion of changes in line 85 and 86 were included in the text. 
  • In line 49, it was deleted 2016 data and keep the tendency about obesity. 
  • Line 53 – during the brewing process of malted barley 
  • It was incorporated a justification for the selection of the products assayed. 
  • EsSense profile was used as a conscious method to explain extrinsic element as was referenced by 28 (Ng et al., 2013). 

Materials and Methods 

  • Brewery spent grain was replaced by BSG in line 90. 
  • The BSG was produced in the same day it was dry, these is included in the manuscripts. 
  • The scientific names of the grains used were included. 
  • Mill model was included in the manuscript. 
  • BSG composition was analysed in Catholic University of Uruguay by authors. 
  • Line 110 was rewritten.  
  • Relative humidity was included in line 132. 
  • Loaves pan was replaced by loaf pans. 
  • In line 136 hs. was replaced by hrs. 
  • We consider that Table 1, 2 and 3 consist in important information for the reproducibility of the work, and should be in the main text. 
  • Line 141 - 000 was deleted from the wheat flour 
  • Line 141 - Particle size was changed 
  • Line 161 - Details for sensory sessions were incorporated. 
  • Line 171 – where was changed for were 
  • Line 175 – a mention to clarify the sentence was added 
  • Line 187 – Title was changed to better suit the content. 
  • 2. Line 188 – a mention to clarify that the samples were three-digit-coded was added 
  • 3. Full questionnaire for pasta, bread and chocolate milk was included in Appendix.  
  • 4. The number of sections and their aims were incorporated.  
  • 5. It was a computer-based questionnaire. This is now detailed. 
  • 6. Line 189 – Explanation included while describing sections of questionnaire. 
  • 7 and 8Line 190 and 191Nine point hedonic scale is a standardized scale which can be reproducible by any researcher, as it shown by Villanueva & Da Silva, 2009Villanueva et al, 2005; Lawless et al., 2010; Nicolas et al., 2010. Wherever, a better explanation of each scale to facilitate understanding was incorporated. 
  • 9. Line 191 – Deleted as no results are reported in this article of this construct. 
  • 10 y 19Line 194 and Line 285– Full form of CATA was deleted, only abbreviation is now provided. 
  • 11. Line 195 – The terms used in CATA questionnaire are presented in full questionnaire shown in Appendix. Better explanation of CATA terms' selection was included in sensory evaluation questionnaire item. 
  • 12. Line 198 – Five point purchase intention scale is a standardized scale which can be reproducible by any researcher. Wherever, a better explanation of each scale used to better understand was incorporated. 
  • 13. Consumers were instructed only to taste the sample before the first section. 
  • 14. Why were the participants’ emotions measured last and only after consumption of the products? The decision of when to test an emotion (before or after product consumption) depends on why you are interested on knowing the participants emotional response. Testing emotions before product consumption might help to identify the feelings which lead to product choice, while testing after gives us and idea of the emotional impact.  
  •    
  • In our case participants didn´t have to make a choice between one product with and another without BSG so our goal was to know the participants emotional impact after product consumption and without prior information of ingredients differences.  Considering this, the EsSense25 was chosen, being a questionnaire developed specifically to assess emotional response in a product testing context; meaning that the participant must first be exposed to the product.   
  •  
  • 15. Table 4 was deleted. Data is shown now in Table 6. 
  • 16. Row 203. Explanation included while describing sections of questionnaire. 
  • 17. Row 206 – Back-translation was done. Detail is now included. 

Results 

  • Scale values of acceptability and purchase intention were incorporated as a foot note in Table 6. 
  • The aim of this work was to study the effect of BSG enrichment in sensory features in different products. We do not focus in obtain a winner product for this enrichment, anyway in Conclusion is written that pasta was the most acceptable product after enrichment 
  • Figure 1, a is mentioned at the end of the sentence. 
  • Line 253. Redaction was check; sentence was edited. 
  • Line 241. Redaction was check; sentence was edited. 
  • Figures 1, 3 and 5 were deleted, results are now presented in new Table (Table 6) 
  • Mention about CATA attribute for each product is important to be detail, because it shows the sensory profile of each product, moreover, of attributes with significance differences. We prefer to keep Figures 2, 4 and 6 to show sensory profile of each product. 
  • Results are now presented in a new Table (Table 7). 
  • Line 330 – Capital letters were replaced   
  • Line 362 – Sentence was corrected. 
  • Line 363 - Sentence was corrected  
  • Line 369 - Reference number was corrected. 
  • Line 418 – Sentence was corrected. 
  • Italic words were changed.   
  • Line 431 – 465 – It was modified the Results and Discussion sections. We consider that for a better understand both sections should be together 
  • Line 485 – Sentence was modified.  
  • Row 515 – WTP means willingness to pay. It was added. 
  • Row 516 – 518 – The concept of top-down modulation was supported in introduction 
  • Line 521 – Pasta spelling was checked. 
  • Line 522 – Sentence was modified. 
  • Ethical statements were incorporated. 
  • A paragraph indicating study limitation was included in the conclusion. 

Reviewer 3 Report

After reading the article, the sensory evaluation basically is complete. However,  it is a primary question that only the two samples evaluated for the sensory evaluation in each session could reduce some applications.

Line 215

McNemar ’s test should be used to analyze for two samples (Cochrran’s Q should be used for three samples or more)

L 331

The results of pasta showed there was no significant difference in the acceptability and purchase intent. However, Line 331 indicated that not buying fiber-enriched pasta is distasteful. I think the author explained this result is rough. I suggested that the result should be discussed carefully through different statistical methods such as PLS regression to understand the effect of sensory and emotional attributes or AHC by panelists.

Line 402 please cite the reference.

Please showed the Table or Figure results of emotional evaluation. It is necessary to understand the relationship between acceptability, purchase intent and emotional  attributes

Author Response

Thank you very much for taking time to review the paper. Manuscript was strongly modified Here are changes we made in order to your comments: 

  • Line 215 - Cochran’s Q Test is a nonparametric way to find differences in matched sets of three or more frequencies or proportions. It is an extension of the McNemar test; the two tests are equal if Cochran’s Q is calculated for two groups McNemar’s test and Cochran’s Q test are equivalent for the two-product case, and both provide a χ2-approximation for conducting pairwise comparisons between products.  

Varela, P. & Ares, G. (2016). Novel Techniques in Sensory Characterization and Consumer Profiling, CRC Press. ISBN 9781138034273 pp.282 

  • Line 331 – Authors agree with the reviewer that a cluster analysis would have provided deeper information about consumers with different patterns in acceptability. However, authors preferred to analyze differences in consumers’ response regarding purchase intention. In fact, reasons to buy or not buy the product are linked to purchase intention results. From all consumers unwilling to buy the product, only 18% of consumer found pasta distasteful. This is a little portion of the total consumers. Perhaps this was not clearly written in the main text. Results on reasons to buy were presented as a table to facilitate understanding of the results and percentage of people who buy or not buy each product was also included in the manuscript. 
  • Line 402 – Reference number 52 is cited. 
  • A table (Table 7) showing results of emotional evaluation was added to facilitate understanding. 

Reviewer 4 Report

Comments are attached.

Author Response

Thank you very much for the positive feedback and for taking time to review the paper carefully. Here are changes we made in order to your comments: 

  • We include BSG definition in abstract. 
  • We change (p<0,05) by (p>0,05) when significant difference was found. 
  • We reexplained sensory evaluation procedures more detailed which include order presentation of samples. 
  • Student’s t test was replaced by Paired t-student test. 
  • We check taste definition and replace taste by flavor. 
  • We change unpleasant aroma and taste by Offlavor. 

Round 2

Reviewer 1 Report

The improvement of the manuscript is evident, particularly in terms of results presented, however, you did not provide chemical compositions of the final products that were specifically addressed as a sincere shortcoming of the manuscript. Although you claim that products possess higher fiber content, you have to confirm this with evident results.

Author Response

Thank you for your comments.  

The nutritional composition tables of each of the products developed are presented as an appendix. 

Reviewer 4 Report

Accept.

Author Response

We attached the last version of the manuscript.

Regards,
